# Outer Nuclear Layer Damage for Detection of Early Retinal Toxicity of Hydroxychloroquine

**DOI:** 10.3390/biomedicines8030054

**Published:** 2020-03-04

**Authors:** Alfonso Casado, Alicia López-de-Eguileta, Soraya Fonseca, Pedro Muñoz, Rosalía Demetrio, Miguel A. Gordo-Vega, Andrea Cerveró

**Affiliations:** Department of Ophthalmology, Hospital Universitario Marqués de Valdecilla-IDIVAL, 39008 Santander, Spain; alicialeguileta@gmail.com (A.L.-d.-E.); sorayafonse@gmail.com (S.F.); pedro.munoz@scsalud.es (P.M.); rosaliademetrio@hotmail.com (R.D.); miguelangel.gordo@scsalud.es (M.A.G.-V.)

**Keywords:** hydroxychloroquine, optical coherence tomography, ganglion cell, outer nuclear layer

## Abstract

In hydroxychloroquine (HCQ) retinopathy, early detection of asymptomatic retinal changes and the interruption of the drug are essential to prevent permanent vision loss. Our purpose was to investigate the roles of ganglion cell layer (GCL) and outer nuclear layer (ONL) thicknesses measured by optical coherence tomography (OCT) in the early diagnosis of retinopathy. One hundred and fourteen eyes of 76 individuals with HCQ treatment were enrolled in the study (42 eyes with impaired visual field (VF) and 72 eyes with nondamaged VF). We found that ONL was significantly decreased in the HCQ retinopathy group compared with the control group in the nasal macula (*p* = 0.032) as well as in four sectors (*p* < 0.044), whereas no significant differences were found comparing GCL in both groups. If VF were altered superiorly or temporarily, ONL was significantly thinned inferiorly (*p* = 0.029) and nasally (*p* = 0.008), respectively. Duration of HCQ treatment was significantly related with ONL in seven sectors of ONL (*p* < 0.047). We suggest that ONL measured with OCT might be used to assess early HCQ retinal toxicity.

## 1. Introduction

Hydroxychloroquine (HCQ) is a 4-aminoquinoline agent and a hydroxylation compound of chloroquine that was first used to prevent or to treat malarial infections. However, the importance of this drug today is the extended use of it in the treatment of plural rheumatic disorders, especially in rheumatoid arthritis (RA) and systemic lupus erythematosus (SLE) [1,2]. Although this drug has deeply proved its benefits [1,2], the prolonged use of HCQ might cause retinal damage as an important side effect, which is known as chloroquine retinopathy, as it was first described with the use of chloroquine [3]. Chloroquine retinopathy has been thoroughly studied and described, focusing on the early detection of retinal damage, as no treatment has yet been proposed. 

Published detection methods of HCQ retinopathy include clinical examinations, baseline color fundus photographs, home Amsler grid testing, and automated perimetry [4]. Corneal chloroquine or HCQ has been related as well with retinopathy, but this theory has not been widely extended [5]. Although full-field electroretinography (ERG) has not been clearly identified as an early marker of HCQ retinopathy [6,7], multifocal ERG might be a reliable device to detect HCQ retinopathy. In fact, the American Academy of Ophthalmology (AAO) recently revised the guidelines for screening, including multifocal electroretinogram, fundus autofluorescence, visual field (VF) examination, and optical coherence tomography (OCT) evaluation [8].

Even though the pathogenesis of HCQ retinopathy has not been completely discerned, in vitro studies have demonstrated that HCQ induced permeability of the retinal pigment epithelial (RPE) layer that contributed to blood–retinal barrier breakdown [9]. Interestingly, with the development of OCT scanning and resolution, it has been possible to in vivo assess the first changes in HCQ-induced toxicity. Retinal layer segmentation allows us to analyze the ganglion cell layer (GCL) together with the inner plexiform layer (IPL) with Cirrus OCT, which is referred to as GCIPL in the literature. Although the first results of this analysis showed no relationship between the thinning of GCIPL and HCQ retinopathy [10], recent results have led other researchers to hypothesize that GCIPL thinning could be an early indicator of it in screening use [11]. However, recent studies have located HCQ damage in the outer retina or the choroidal tissue, which is close to the findings of in vitro studies [12,13,14,15]. Some of these studies assessed the retina using OCT angiography. This device allowed them to obtain en face images that could be analyzed by two investigators in a masked manner [13,15]. However, OCT segmentation could depict objective values of thickness of retinal layers. Thus, the aim of this study was to assess the thickness of the adjacent RPE layer:outer nuclear layer (ONL) and the distant RPE layer:GCL using posterior pole analysis (PPA) in order to find an objective indicator of early HCQ retinal toxicity. 

## 2. Materials and Methods

### 2.1. Individuals

All individuals were recruited from the ophthalmology department of Marqués de Valdecilla University Hospital, from May 2016 to June 2018. The study protocol was approved by the Ethics Committee of Marqués de Valdecilla University Hospital (approved on 12 July 2017 and code: 2017.146), and it was performed in accordance with the tenets of the Declaration of Helsinki. Written consent forms were signed by all the participants before the examinations.

All individuals enrolled had a refractive error less than +6.0 diopters (DP) and more of −6 DP of sphere or 3 diopters of cylinder and no history of retinal diseases (for instance, macular degeneration, central serous chorioretinopathy, or diabetic retinopathy). Individuals with objective photoreceptor disruption in OCT, such as missing, poorly reflective, or interruption of the ellipsoid band, as well as the potentially representing misaligned photoreceptor, were excluded. One hundred twenty-four eyes from 81 individuals (75 females and 6 males) that received HCQ were included in the study. Age, sex, diagnosis, the daily dose of HCQ, and duration of use were recorded for all individuals. Since retinal layers’ thicknesses are related with glaucoma and other various retinal pathologies, we excluded all eyes with ocular diseases, as well as clinically relevant opacities of the optic media and low-quality images due to unstable fixation, or severe cataract (individuals with mild to moderate cataract could be enrolled in the study, but only high-quality images were included). If the fellow eye was used in this study, only the affected eye was removed from the study, but the patient was not. Patients in treatment with HCQ with hepatic or renal failure that might increase retinal toxicity or patients in treatment with medicines which cause retinal toxicity were also excluded from the study. Subsequently, 10 eyes were excluded due to errors in the retinal layer segmentation with the OCT, so 114 eyes of 76 individuals were finally analyzed in this study. 

### 2.2. Clinical Assessment

All individuals underwent an exhaustive ophthalmic exploration on the day of OCT imaging, consisting of the best-corrected visual acuity, refraction, slit lamp examination, fundus examination, OCT examination, and intraocular pressure (IOP) measurement with Goldmann applanation tonometry (in this order). The refractive error was analyzed using an autorefractometer (Canon RK-F1, Canon USA Inc., Lake Success, NY, USA). 

### 2.3. Optical Coherence Tomography Procedure 

#### Methodology for Measurement of ONL and GCL

A single, well-prepared ophthalmologist (A.L.E.) conducted all the OCT measurements. The retinal thickness was examined with the Spectralis OCT (Heidelberg Engineering, Heidelberg, Germany) using the images obtained by PPA scans. Employing this protocol, this device automatically defined a line connecting the center of the optic disc and the center of the fovea as a reference line. Afterwards, 61 line scans (1024 A scans/line) parallel to the central reference line were delineated. The quality of these scans was indicated with a color scale at the bottom of the analyzed images. Only if this was in the green range was it considered a good quality scan and included in this study. The quality of the exams was 28.2 ± 2.3 and 27.8 ± 2.2 in patients with or without VF defects, respectively. A masked investigator (S.F.) examined all of the images of each eye to determine whether there was any segmentation or centered errors in the images. Ten eyes were excluded because of a segmentation mistake in the retinal layers. The average retinal layer measurement of each 8 × 8 (3° × 3°) sector, which made up the 64 sectors, was settled. As HCQ toxicity is located in the center of the macula [12,13,14], only 4 × 4 central grids were analyzed to expedite the study. These 16 sectors were numbered as shown in Figure 1. The nasal sector included the numbered 3, 4, 7, 8, 11, 12, 15, and 16 sectors; the temporal sector included 1, 2, 5, 6, 9, 10, 13, and 14; and superior from 1 to 8, whereas inferior from 9 to 16 sectors.

### 2.4. Visual Field

A 10-2 VF (Humphrey Field test program) was performed in all patients included in this study. VF damage was classified according to the impairment sector (nasal, temporal, superior, and inferior) as shown in Figure 2. VF was classified as normal when there was a mean deviation or a pattern standard deviation within the 95th percentile.

### 2.5. Statistical Analysis 

All statistical analyses were performed using IBM SPSS Statistics V.20.0 (International Business Machine Corporation, Armonk, NY, USA). 

A one-sample Kolmogorov–Smirnov test was used to verify the normality of the data distribution. As ONL in the temporal region did not follow a normal distribution, nonparametric tests were used. The Mann–Whitney test was used to compare ONL and GCL between eyes with VF damage and control eyes. The Kruskal–Wallis test was used to assess differences between sectors for each VF defect. A post hoc Tukey’s test was used to compare each sector. A Spearman correlation test was used to analyze the relationship between retinal layers and duration of HCQ treatment. The level of statistical significance was set at *p* < 0.05.

## 3. Results 

Overall, 114 eyes of 76 individuals (71 females and 5 males) were included in the study. Those five males were both in the control group (three patients) and in the patient group (two controls). The mean age of the individuals was 58.1 ± 14.2 years (age range: 19-83 years). The mean spherical equivalent was measured as +0.56 ± 0.41 diopters. Mean IOP was 13.6 ± 2.1 mmHg (IOP range: 10–19 mmHg). Mean duration of HCQ treatment was 41.3 months. Fifty-five individuals (72%) were treated for SLE and 21 (28%) for RA. Table 1 shows the demographic and clinical participants’ characteristics, comparing control and HCQ retinopathy eyes. Individuals with impaired VF were considered as early HCQ retinopathy.

Table 2 and Table 3 show the differences between control and eyes with damaged VF due to HCQ regarding the analysis of the ONL and the GCL, respectively. Interestingly, eyes with HCQ retinopathy were associated with a significant thinning of ONL in the nasal group (*p* = 0.032) as well as in sectors 11, 12, 15, and 16 (*p* = 0.009, *p* = 0.017, *p* = 0.044, and *p* = 0.014, respectively). No significant differences were depicted in GCL thicknesses between HCQ retinopathy eyes and control eyes (*p* > 0.101).

Table 4 shows the values of ONL depending on the sector of VF damage using the Kruskal–Wallis test. Eyes with superior damage were associated with a lower value of ONL in the inferior region (*p* = 0.029), and eyes with temporal impairment were associated with a lower value of ONL in the nasal region (*p* = 0.008). The post hoc Tukey’s test confirmed that, in patients with superior VF defect, inferior ONL thinning was significant compared with nasal, temporal, or superior ONL values (*p* < 0.001). Similarly, in patients with temporal VF defect, nasal ONL thinning was significant compared with temporal, superior, or inferior ONL values (*p* < 0.001). 

Interestingly, the Spearman correlation test showed that the duration of HCQ treatment was significantly related with ONL in sectors 1, 5, 9, 10, 11, 13, and 14 (*p* = 0.046, *p* = 0.017, *p* = 0.014, *p* = 0.003, *p* = 0.047, *p* = 0.015, *p* = 0.024, and *p* = 0.046, respectively). No sector of GCL was correlated with the duration of HCQ treatment (*p* > 0.211).

## 4. Discussion

The efficacy of HCQ in preventing RA or SLE flares has been well illustrated [16,17,18,19]. Evidence sufficiently supports the use of HCQ in SLE, as it has been proved that HCQ improves the disease activity and allows the reduction of the corticosteroid dose regardless of background treatment [16], as well as that the adherence to HCQ improves long-term survival [17]. Moreover, the drug has a low cost and few side effects. Thus, rheumatoid experts encourage greater use of HCQ, also when organ involvement is found [18]. In RA as well, it was recently published that the clinical and structural efficacy of HCQ was similar to that for methotrexate or sulfasalazine in monotherapy, with fewer adverse effects [19]. However, one of its main side effects is HCQ retinopathy. This finding was first described in 1959 in patients that used chloroquine [3], and since then, HCQ retinopathy has been well documented in several papers [4,5,6,7,8,9,10,11,12,13,14,15], as early detection of it and drug discontinuation prevents the patient from developing established visual symptoms [13,14,15]. It is important to point out that the discontinuation of hydroxychloroquine does not prevent further macular deterioration. However, the cessation of the drug might decrease the rate of deterioration. Thus, the early detection of hydroxychloroquine damage could be relevant and it is available today, as the development of OCT provides a sufficient amount of information on the retina so that the first changes that HCQ yields in this tissue might be studied [13,14,15]. 

Previous reports positioned this opening damage in the outer retina or the choroidal tissue, just as in vitro studies’ have proposed [14,15]. As retinal segmentation is available, our study examined different previously proposed retinal layers in the early cases of HCQ retinopathy to establish well the first damaged layer in these cases. Hence, the PPA protocol was used in these patients as this analysis. PPA provides the investigator 64 sector thicknesses in a 3° × 3° macular region, and it has been shown to be a reliable test in glaucoma [20]. Since it provides a large amount of information, it could be a good test to locate early sector damage in HCQ retinopathy. The interobserver and intervisit reproducibility of the PPA has been reported to be excellent [21,22]. One of these proposed damaged layers was the GCL. Although this layer has been well studied and proven its damage in glaucoma [23], Alzheimer’s disease [24], and Parkinson’s disease [25], we found no significant differences in eyes with early HCQ retinopathy and control eyes. As the GCL is connected with brain tissues, it is plausible that in neurological diseases this could be the first affected retinal layer. In fact, even plaques of extracellular Aβ deposits have been found in the GCL in AD patients’ retina [26]. However, in vitro studies suggested that the accumulation of HCQ was located in the outer retina or the choroidal tissue. Thus, our results might solve the previous discrepancies of some studies that found GCL thinning in HCQ retinopathy [11] and others that did not find any significant change in this layer [10]. On the other hand, our findings support that the first damaged retinal layer might be the ONL, as we discovered a significant thinning in patients with early HCQ retinopathy compared with control eyes in different retinal sectors, especially in the nasal area, as well as a significant correlation of ONL thickness and the duration of HCQ treatment. Thus, we suggest this layer to be considered as a promising early detector of HCQ retinopathy and we recommend focusing on this examination to discern early cases. 

The AAO has suggested screening recommendations for HCQ retinopathy [1,2]. The aim of this screening is to recognize clear signs of toxicity at an early enough stage to prevent vision loss for this drug. However, new OCT tools such as OCT angiography and choroidal thickness might sound promising. Hence, OCT angiography has also been suggested as a good biomarker of HCQ retinopathy. Bulut et al. [14] studied 60 eyes and found a significant wide foveal avascular zone (FAZ) in patients with high-risk HCQ retinopathy. However, this study did not differentiate HCQ retinopathy patients according to the damage of the VF but used the duration of HCQ treatment to discriminate the risk of HCQ retinopathy. Differently, we classified patients depending on their VF. Our findings revealed an association of the thinned hemiretina with the opposite hemifield, especially in abnormal superior and temporal damaged VF, where we found a significant thinned inferior and nasal ONL, respectively. Similarly, choroidal thickness has also been suggested to be a good predictor of HCQ retinopathy. Ahn et al. [15] reported significant choroidal thinning in eyes with HCQ retinopathy. Temporal and nasal 0.5–3 mm measurements were made from the fovea. They found significant differences in every nasal location but no significant differences in the temporal choroid at 1.5 mm from the fovea. Interestingly, our results also showed significant thinning in nasal sectors of the ONL. Thus, we hypothesize that HCQ retinopathy might begin in the nasal region of the macula. This fact might be related with the abundance of orbital vessels, many of which enter the choroid posteriorly and nasally, controlled by vasoactive intestinal polypeptide and neuronal nitric oxide synthase, the fibers of which are most abundant in the nasal choroid, as recently published [27]. These vasoactive substances might promote HCQ extravasation in the nasal region of the macula and choroidal tissues, and consequently, the first observable changes in HCQ retinopathy could be in the nasal hemimacula. 

Several limitations are present in this study. First, the diagnosis of HCQ retinopathy was made using VF, OCT, and funduscopy, but neither multifocal electroretinogram nor fundus autofluorescence were performed. However, we hypothesize that ONL thinning is observable before electroretinogram or autofluorescence changes are observed. Secondly, we assessed the changes only in 16 sectors of the macula. However, this might represent the area where HCQ retinopathy is located and we chose these sectors to develop an easily feasible early detection system. Finally, there is no study that previously analyzed ONL in patients with HCQ retinopathy, and thus, the results of our study could not be compared.

In conclusion, this study found an association between ONL thinning and HCQ retinopathy and used a proved and objective analysis (PPA) to assess these changes. As it has not been previously reported, changes in the ONL might be studied in further studies of larger numbers of patients with HCQ retinopathy in order to correctly stablish this layer as a useful tool in the early detection of HCQ retinopathy, to detect this adverse event at the stage when it is still reversible.

## Figures and Tables

**Figure 1 biomedicines-08-00054-f001:**
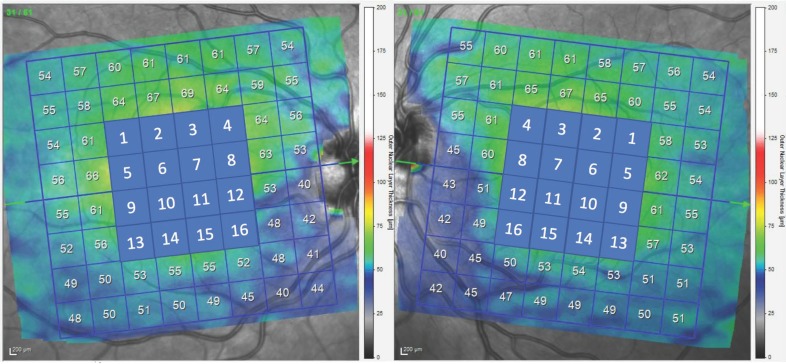
OCT scan of the macula with the posterior pole analysis (PPA) showing the 16 sectors numbered from 1 to 16. The nasal sector included the numbered 3, 4, 7, 8, 11, 12, 15, and 16 sectors; the temporal sector included 1, 2, 5, 6, 9, 10, 13, and 14; and superior from 1 to 8, whereas inferior from 9 to 16 sectors. Green arrows represent the horizontal line centred in fovea.

**Figure 2 biomedicines-08-00054-f002:**
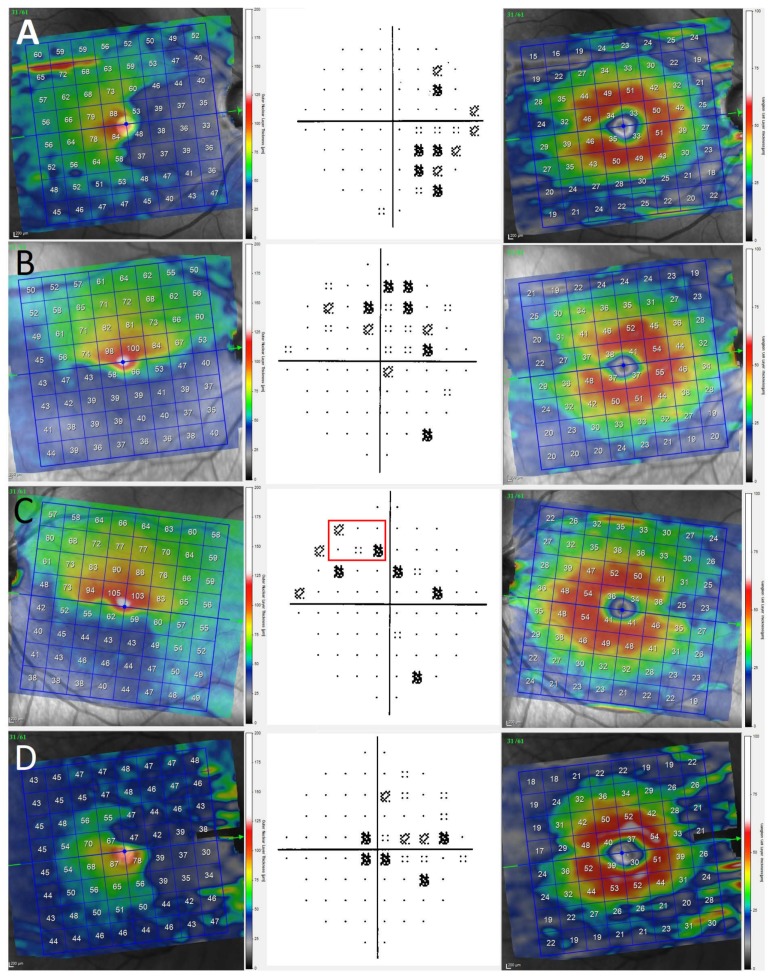
Three different patients (**A**–**D**) with HCQ retinopathy with their visual fields and OCT, showing a perfect correspondence of both damage tests. Left images show the outer nuclear layer (ONL) analysis, center images the visual fields (VFs), and right images represent ganglion cell layer (GCL) analysis. (**A**) right eye of a 65-year-old woman with systemic lupus erythematosus that presented with nasal damage of ONL-associated temporal damage of VF with normal GCL analysis; (**B**,**C**) right and left eyes of a 71-year-old woman with RA showing inferior impairment of ONL and superior VF damage, keeping a nonaltered GCL measurement; (**D**) 53-year-old man with systemic lupus erythematosus and nasal damage of ONL with temporal damage of VF preserving normal GCL analysis. Green arrows represent the horizontal line centred in fovea. The symbols in the red box represents de located damage in visual field.

**Table 1 biomedicines-08-00054-t001:** Demographic and clinical participants’ characteristics of hydroxychloroquine (HCQ) retinopathy eyes and control eyes (114 eyes of 76 individuals). Data for quantitative variables are shown as mean (standard deviation). Intraocular pressure (IOP). Analysis was performed using the Mann–Whitney test.

	HCQ Retinopathy Eyes (*n* = 42)	Control Eyes (*n* = 72)	*p*
Age (years)	63.1 (14.9)	53.8 (13.5)	0.118
Female eyes (%)	38 (90.4)	66 (91.7)	0.906
Spherical equivalent (diopters)	0.55 (0.39)	0.67 (0.44)	0.144
IOP	13.9 (2.4)	13.4 (2.1)	0.361

**Table 2 biomedicines-08-00054-t002:** Comparison of outer nuclear layer (ONL) between hydroxychloroquine retinopathy (HCQR) eyes and control eyes. Data for quantitative variables are shown as mean (standard deviation). Analysis was performed using the Mann–Whitney test. * *p*-value is <0.05.

	HCQR Eyes (*n* = 42)	Control Eyes (*n* = 72)	*p*
ONL 1	62.5 (6.7)	62.9 (7.3)	0.473
ONL 2	65.3 (9.6)	67.0 (8.2)	0.173
ONL 3	63.9 (11.5)	66.4 (8.2)	0.177
ONL 4	60.4 (9.5)	62.8 (8.2)	0.111
ONL 5	70.6 (8.5)	70.7 (8.9)	0.920
ONL 6	78.7 (13.4)	78.5 (11.7)	0.947
ONL 7	73.9 (15.6)	75.2 (14.3)	0.507
ONL 8	63.7 (14.1)	75.2 (14.7)	0.080
ONL 9	67.2 (12.0)	68.3 (8.8)	0.664
ONL 10	76.1 (13.9)	79.7 (12.6)	0.357
ONL 11	71.2 (17.5)	80.2 (13.3)	0.009 *
ONL 12	61.5 (14.6)	68.5 (12.1)	0.017 *
ONL 13	56.9 (8.3)	59.8 (6.7)	0.279
ONL 14	58.5 (11.4)	63.2 (8.3)	0.101
ONL 15	66.5 (6.8)	62.4 (9.5)	0.044 *
ONL 16	53.6 (10.3)	58.5 (8.8)	0.014 *
Superior ONL	539.3 (73.3)	551.2 (65.3)	0.161
Inferior ONL	511.5 (96.4)	540.5 (68.3)	0.104
Temporal ONL	535.9 (58.2)	550.1 (59.5)	0.264
Nasal ONL	514.9 (93.1)	541.6 (71.7)	0.032 *
Total ONL	1050.8 (113.8)	1091.7 (117.4)	0.072

**Table 3 biomedicines-08-00054-t003:** Comparison of ganglion cell (GC) layer between hydroxychloroquine retinopathy (HCQR) eyes and control eyes. Data for quantitative variables are shown as mean (standard deviation). Analysis was performed using the Mann–Whitney test.

	HCQR Eyes (*n* = 42)	Control Eyes (*n* = 72)	*p*
GC 1	41.9 (4.7)	43.4 (5.1)	0.121
GC 2	49.3 (5.3)	50.7 (5.5)	0.176
GC 3	49.9 (5.5)	50.5 (5.6)	0.478
GC 4	43.4 (5.3)	43.5 (5.0)	0.889
GC 5	42.2 (7.5)	45.9 (6.1)	0.113
GC 6	32.9 (5.6)	33.3 (6.1)	0.730
GC 7	34.7 (6.1)	35.5 (6.3)	0.398
GC 8	51.3 (6.5)	52.7 (6.1)	0.241
GC 9	47.3 (7.0)	49.9 (6.4)	0.157
GC 10	35.1 (6.3)	35.8 (6.2)	0.545
GC 11	34.9 (6.4)	35.2 (6.4)	0.754
GC 12	51.5 (6.4)	53.3 (5.9)	0.155
GC 13	42.3 (4.7)	43.8 (4.6)	0.141
GC 14	48.9 (4.7)	50.3 (4.7)	0.101
GC 15	48.5 (5.0)	50.1 (4.8)	0.135
GC 16	43.3 (5.2)	44.3 (4.6)	0.368
Superior GC	345.6 (39.9)	355.4 (37.3)	0.231
Inferior GC	351.8 (36.3)	362.6 (36.3)	0.194
Temporal GC	339.9 (37.8)	352.9 (36.3)	0.188
Nasal GC	357.5 (38.8)	365.1 (36.5)	0.303
Total GC	697.4 (74.9)	718.1 (71.5)	0.180

**Table 4 biomedicines-08-00054-t004:** Comparison of the outer nuclear layer (ONL) and visual field defect (D) in hydroxychloroquine retinopathy (HCQR) eyes. Data for quantitative variables are shown as mean (standard deviation). Analysis was performed using the Kruskal–Wallis test. * *p*-value is <0.05.

	Superior ONL	Inferior ONL	Temporal ONL	Nasal ONL	*p*
Inferior D	512.4 (28.9)	620.7 (75.9)	532.7 (48.1)	600.4 (12.4)	0.347
Superior D	556.5 (98.2)	474.6 (92.7)	541.7 (66.5)	516.4 (75.3)	0.029 *
Nasal D	618.0 (46.7)	553.0 (38.2)	604.5 (58.7)	566.5 (26.2)	0.125
Temporal D	527.2 (57.3)	469.2 (51.4)	550.4 (47.6)	446.0 (49.8)	0.008 *

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
