# Peer review of "Outer Nuclear Layer Damage for Detection of Early Retinal Toxicity of Hydroxychloroquine"

_biomedicines, 2020, doi:10.3390/biomedicines8030054_

Round 1
Reviewer 1 Report
Retinopathy is a common and serious complication to the use of HCQ. Discontinuing the drug in the early stages can prevent permanent vision loss. Therefore, early detection of asymptomatic retinal changes is critical to avoid irreversible damage. In this study, the authors propose that nasal thinning of the ONL and corresponding VF impairment are among the earliest asymptomatic changes and confirm that GCL was not affected in HCQ retinopathy. The introduction needs to be improved and the rational for this work needs to be described better.
Author Response
We want to thank the reviewers for the speed of their responses. All changes are highlighted in red in the attached document.
REVIEWER 1
Retinopathy is a common and serious complication to the use of HCQ. Discontinuing the drug in the early stages can prevent permanent vision loss. Therefore, early detection of asymptomatic retinal changes is critical to avoid irreversible damage. In this study, the authors propose that nasal thinning of the ONL and corresponding VF impairment are among the earliest asymptomatic changes and confirm that GCL was not affected in HCQ retinopathy. The introduction needs to be improved and the rational for this work needs to be described better.
We agree with the reviewer, so we have rewritten the introduction as follows (Page 1, Lines 26-33):
Hydroxychloroquine (HCQ) is a 4-aminoquinoline agent and a hydroxylation compound of chloroquine that has been first used to prevent or to treat malarial infections. However, the importance of this drug today is the extended use of it in the treatment of plural rheumatic disorders, especially in rheumatoid arthritis (RA) and systemic lupus erythematosus (SLE) (1, 2). Although this drug has deeply proved its benefits (1,2), the prolonged use of HCQ might cause retinal damage as an important side effect, which is known as chloroquine retinopathy, as it was first described with the use of chloroquine (3). Chloroquine retinopathy has been thoroughly studied and described, focusing in an early detection of retinal damage, as no treatment have already been proposed.
As well as the methods section (Pages 2-3, lines 38-5):
2.2. Clinical assessment
All individuals underwent an exhaustive ophthalmic exploration on the day of OCT imaging, consisting of the best-corrected visual acuity, refraction, slit lamp examination, fundus examination, OCT examination and intraocular pressure (IOP) measurement with goldmann applanation tonometry (in this order). The refractive error was analysed using an auto refractometer (Canon RK-F1, Canon USA Inc., Lake Success, NY, USA).
2.3. Optical coherence tomography procedure
Methodology for measurement of ONL and GCL:
A single, well-prepared ophthalmologist (ALE) conducted all the OCT measurements. The retinal thickness was examined with the Spectralis OCT (Heidelberg Engineering, Heidelberg, Germany) using the images obtained by PPA scans. Employing this protocol, this device automatically define a line connecting the center of the optic disc and the center of the fovea as a reference line. Afterwards, 61 line scans (1024 A scans/line) parallel to the central reference line were delineated. The quality of these scans is indicated with a colour scale at the bottom of the analysed images.
Reviewer 2 Report
The authors of the manuscript entitled "Outer nuclear layer damage for detection of early retinal toxicity of hydroxychloroquine" have performed OCT scans in 114 eyes belonging to 76 individuals.
The work is well written although the result section should be improved to facilitate the reading. The same occurs with the abstract. The abstract is divided into sections, while the journal publishes the abstracts as one single paragraph. Please arrange accordingly.
COMMENTS:
Please indicate Figure and Table numbers. Right now the text does not have any figure number. The authors should explain better the groups and how they were distributed. It is quite confusing that initially, they had 124 eyes from 81 patients. Then 10 eyes were discarded (I assume from 5 patients) to later indicate that there were 114 eyes from 76 patients. But the overall distribution (also male-female) is not really clear prior to the exclusion. Following in the comment above, the end of the study is also not known (started in May 2016 to June 2016?) Another confusing point is that the authors state that all 114 eyes were from "patients" however they have 42 HCQ retinopathy eyes and 72 control. I would suggest that the authors clearly use the word individuals to refer to the entire group, and they differentiate between control and patients. One noticeable aspect is that only 5 males were included. Can the authors elaborate on that? where they in the control group or in the patient one? Why only 5? is the HCQ retinopathy more prevalent in females? Revise numbers on table 1. this reviewer thinks that the female % may not be correct (taking into account that only a maximum of 10 eyes might be from males, it is impossible a 36% and 66% in each group. In addition, if this is the number of eyes it should be indicated and then 66/72 is not 8.3%. Please revise the entire table. Table 2 should be split in 2. Results of ONL and results of GC, otherwise it is confusing because of the total numbers. Statistics, the authors have uses Mann-Whitney test. Could they elaborate on why? Taking into account the number of measurements made, this reviewer thinks that other statistical tests should have been used. As long there is a clear reason behind it.
MINOR COMMENTS
Rephrase sentence in line 5 page 2. It is weird that they indicate that the authors hypothesized that the GC-IPL thinning could be an early indicator, but then the authors refer to a paper that does not belong to them. I assume they aimed to say that other researchers... also in the way the sentence is written it should read "made the authors hypothesize" (without the d). The degree symbol is not correct. Last but not least, it is surprising that authors do not have any financial support (not even internal) nor acknowledgments.
Author Response
The authors of the manuscript entitled "Outer nuclear layer damage for detection of early retinal toxicity of hydroxychloroquine" have performed OCT scans in 114 eyes belonging to 76 individuals.
The work is well written although the result section should be improved to facilitate the reading. The same occurs with the abstract. The abstract is divided into sections, while the journal publishes the abstracts as one single paragraph. Please arrange accordingly.
We agree with the reviewer, so we have changed the abstract as follows (Page 1, lines 10-22):
In hydroxychloroquine (HCQ) retinopathy, early detection of asymptomatic retinal changes and the interruption of the drug is essential to prevent permanent vision loss. Our purpose was to investigate the role of ganglion cell layer (GCL) and outer nuclear layer (ONL) thicknesses measured by optical coherence tomography (OCT) in the early diagnosis of retinopathy. One hundred fourteen eyes of 76 individuals with HCQ treatment were enrolled in the study (42 eyes with impaired visual field (VF) and 72 eyes with non-damaged VF). We found that ONL was significantly decreased in the HCQ retinopathy group compared to the control group in the nasal macula (p=0.032) as well as in 4 sectors (p<0.044), whereas no significant differences were found comparing GCL in both groups. If VF were altered superiorly o temporarily, ONL was significantly thinned inferiorly (p=0.029) and nasally (p=0.008) respectively. Duration of HCQ treatment was significantly related with ONL in 7 sectors of ONL (p<0.047). We suggest that ONL measured with OCT might be used to assess early HCQ retinal toxicity.
As well as the results section ( Page 5, lines 11-13):
“Overall, 114 eyes of 76 individuals (71 females and 5 males) were included in the study. Those 5 males were both in the control group (3 patients) and in the patient group (2 controls). The mean age of the individuals was 58.1 ± 14.2 years (age range: 19-83 years).”
COMMENTS:
Please indicate Figure and Table numbers. Right now the text does not have any figure number.
We have added the number of each figure in each corresponding paragraph.
Page 3, line 11-12: These 16 sectors were numbered as shown in Figure 1.
Page 3, line 15 (figure caption): Figure 1.
Page 3, line 20-21: VF damaged were classified according to the impairment sector (nasal, temporal, superior, inferior) as showed in Figure 2.
Page 4: line 2 (figure caption): Figure 2.
Page 5, line 20. Table 1.
Page 5, line 29: Table 2.
Page 6, line 1: Table 3
Page 7, line 9: Table 4.
The authors should explain better the groups and how they were distributed. It is quite confusing that initially, they had 124 eyes from 81 patients. Then 10 eyes were discarded (I assume from 5 patients) to later indicate that there were 114 eyes from 76 patients. But the overall distribution (also male-female) is not really clear prior to the exclusion. Following in the comment above, the end of the study is also not known (started in May 2016 to June 2016?)
Another confusing point is that the authors state that all 114 eyes were from "patients" however they have 42 HCQ retinopathy eyes and 72 control. I would suggest that the authors clearly use the word individuals to refer to the entire group, and they differentiate between control and patients.
We agree with the reviewer so we have clarified the number and sex of the individuals that were included in the study, using the term individuals to refer to the entire group, as advised by the reviewer:
Page 2, lines 27-28: One hundred twenty four eyes from 81 individuals (75 females and 6 males) that received HCQ were included in the study.
Page 2, lines 36-37: Subsequently, 10 eyes (4 females and 1 male) were excluded due to errors in the retinal layer segmentation with the OCT, so 114 eyes of 76 individuals were finally analyzed in this study.
Page 5, lines 12-13: Overall, 114 eyes of 76 individuals (71 females and 5 males) were included in the study. The mean age of the individuals was 58.1 ± 14.2 years.
Page 5 line 16: 55 individuals (72%) were treated for SLE and 21 (28%) for RA.
Page 5, lines 18-19: Individuals with impaired VF were considered as early HCQ retinopathy.
As well in individuals section:
2.1. Individuals (Page 2, lines 17-37)
All individuals were recruited from the ophthalmology department of Marqués de Valdecilla University Hospital, from May 2016 to June 2018. The study protocol was approved by the Ethics Committee of Marqués de Valdecilla University Hospital, and it was performed in accordance with the tenets of the Declaration of Helsinki. Written consent forms were signed from all the participants before the examinations.
All individuals enrolled had a refractive error less than +6.0 diopters (DP) and more of -6 DP of sphere or 3 diopters of cylinder, no history of retinal diseases (for instance: macular degeneration, central serous chorioretinopathy, diabetic retinopathy). Individuals with objective photoreceptor disruption in OCT, such as missing, poorly reflective or interruption of the ellipsoid band, as well as the potentially representing misaligned photoreceptor, were excluded. One hundred twenty four eyes from 81 individuals (75 females and 6 males) that received HCQ were included in the study. Age, sex, diagnosis, the daily dose of HCQ, and duration of use were recorded for all individuals. Since retinal layers’ thicknesses are related with glaucoma and other various retinal pathologies, we excluded all individuals with ocular diseases, as well as clinically relevant opacities of the optic media and low-quality images due to unstable fixation, or severe cataract (individuals with mild to moderate cataract could be enrolled in the study, but only high-quality images were included). Patients in treatment with HCQ with hepatic or renal failure that might increase retinal toxicity or patients in treatment with medicines which cause retinal toxicity were also excluded from the study. Subsequently, 10 eyes (5 females and 1 male) were excluded due to errors in the retinal layer segmentation with the OCT, so 114 eyes of 76 individuals were finally analyzed in this study.
On the other hand, we have completed the dates of the end of the study:
Page 2, line 19: from May 2016 to June 2018.
One noticeable aspect is that only 5 males were included. Can the authors elaborate on that? where they in the control group or in the patient one? Why only 5? is the HCQ retinopathy more prevalent in females?
As the reviewer comments, it is true that the number of males is quite less than the number of females. This is probably since the rheumatic diseases in which hydroxychloroquine treatment is used (Lupus, rheumatoid arthritis, ...) are much more frequent in females than in males (Yoo DH, Suh CH, Shim SC, Jeka S, Cons-Molina FF, Hrycaj P, et al. A multicentre randomised controlled trial to compare the pharmacokinetics, efficacy and safety of CT-P10 and innovator rituximab in patients with rheumatoid arthritis. Ann Rheum Dis. 2017 Mar;76(3):566-570.). Similarly, HCQ is much more frequent in females. For instance, in reference 12 (de Sisternes L, Hu J, Rubin DL, Marmor MF. Localization of damage in progressive hydroxychloroquine retinopathy on and off the drug: inner versus outer retina, parafovea versus peripheral fovea. Invest Ophthalmol Vis Sci 2015;56(5):3415–3426.) de Sisternes et al included 11 patients with HCQ, and 10 of them were females.
In reference to the group in which the 5 males included in the study are located, three males were in the control group (non-damaged VF) and the other two were in the patient group (damaged VF). Therefore, we have added in the manuscript:
“Those 5 males were both in the control group (3 eyes) and in the patient group (2 eyes).” (page 5, lines 12-13)
Revise numbers on table 1. this reviewer thinks that the female % may not be correct (taking into account that only a maximum of 10 eyes might be from males, it is impossible a 36% and 66% in each group. In addition, if this is the number of eyes it should be indicated and then 66/72 is not 8.3%. Please revise the entire table.
We regret the confusion, the number of male eyes between control eyes is 6, so the number of female eyes is 66/72 (91.7%). Similarly, the number of male eyes between HCQ retinopathy eyes is 4, so the female eye number is 38/42 (90.4%). We have corrected the values in the table as follows (Table 1):
|
|
HCQ retinopathy eyes (N=42) |
Control eyes (N=72) |
P |
|
Age (years) |
63.1 (14.9) |
53.8 (13.5) |
0.118 |
|
Female eyes (%) |
38 (90.4) |
66 (91.7) |
0.906 |
|
Spherical equivalent (Diopters) |
0.55 (0.39) |
0.67 (0.44) |
0.144 |
|
IOP |
13.9 (2.4) |
13.4 (2.1) |
0.361 |
Table 2 should be split in 2. Results of ONL and results of GC, otherwise it is confusing because of the total numbers.
As the reviewer recommends we have divided Table 2 into two tables (Table 2 and Table 3).
(Page 5, Line 29)
Table 2. Comparison of Outer Nuclear Layer (ONL) between hydroxychloroquine retinopathy (HCQR) eyes and control eyes. Data for quantitative variables are shown as mean (standard deviation). Analysis was performed using Mann-Whitney test. * p value is < 0.05.
|
|
HCQR eyes (N=42) |
Control eyes (N=72) |
P |
|
ONL 1 |
62.5 (6.7) |
62.9 (7.3) |
0.473 |
|
ONL 2 |
65.3 (9.6) |
67.0 (8.2) |
0.173 |
|
ONL 3 |
63.9 (11.5) |
66.4 (8.2) |
0.177 |
|
ONL 4 |
60.4 (9.5) |
62.8 (8.2) |
0.111 |
|
ONL 5 |
70.6 (8.5) |
70.7 (8.9) |
0.920 |
|
ONL 6 |
78.7 (13.4) |
78.5 (11.7) |
0.947 |
|
ONL 7 |
73.9 (15.6) |
75.2 (14.3) |
0.507 |
|
ONL 8 |
63.7 (14.1) |
75.2 (14.7) |
0.080 |
|
ONL 9 |
67.2 (12.0) |
68.3 (8.8) |
0.664 |
|
ONL 10 |
76.1 (13.9) |
79.7 (12.6) |
0.357 |
|
ONL 11 |
71.2 (17.5) |
80.2 (13.3) |
0.009* |
|
ONL 12 |
61.5 (14.6) |
68.5 (12.1) |
0.017* |
|
ONL 13 |
56.9 (8.3) |
59.8 (6.7) |
0.279 |
|
ONL 14 |
58.5 (11.4) |
63.2 (8.3) |
0.101 |
|
ONL 15 |
66.5 (6.8) |
62.4 (9.5) |
0.044* |
|
ONL 16 |
53.6 (10.3) |
58.5 (8.8) |
0.014* |
|
Superior ONL |
539.3 (73.3) |
551.2 (65.3) |
0.161 |
|
Inferior ONL |
511.5 (96.4) |
540.5 (68.3) |
0.104 |
|
Temporal ONL |
535.9 (58.2) |
550.1 (59.5) |
0.264 |
|
Nasal ONL |
514.9 (93.1) |
541.6 (71.7) |
0.032* |
|
Total ONL |
1050.8 (113.8) |
1091.7 (117.4) |
0.072 |
(Page 6, line 1)
Table 3. Comparison of Ganglion Cell Layer (GC) between hydroxychloroquine retinopathy (HCQR) eyes and control eyes. Data for quantitative variables are shown as mean (standard deviation). Analysis was performed using Mann-Whitney test. * p value is < 0.05.
|
|
HCQR eyes (N=42) |
Control eyes (N=72) |
P |
|
GC 1 |
41.9 (4.7) |
43.4 (5.1) |
0.121 |
|
GC 2 |
49.3 (5.3) |
50.7 (5.5) |
0.176 |
|
GC 3 |
49.9 (5.5) |
50.5 (5.6) |
0.478 |
|
GC 4 |
43.4 (5.3) |
43.5 (5.0) |
0.889 |
|
GC 5 |
42.2 (7.5) |
45.9 (6.1) |
0.113 |
|
GC 6 |
32.9 (5.6) |
33.3 (6.1) |
0.730 |
|
GC 7 |
34.7 (6.1) |
35.5 (6.3) |
0.398 |
|
GC 8 |
51.3 (6.5) |
52.7 (6.1) |
0.241 |
|
GC 9 |
47.3 (7.0) |
49.9 (6.4) |
0.157 |
|
GC 10 |
35.1 (6.3) |
35.8 (6.2) |
0.545 |
|
GC 11 |
34.9 (6.4) |
35.2 (6.4) |
0.754 |
|
GC 12 |
51.5 (6.4) |
53.3 (5.9) |
0.155 |
|
GC 13 |
42.3 (4.7) |
43.8 (4.6) |
0.141 |
|
GC 14 |
48.9 (4.7) |
50.3 (4.7) |
0.101 |
|
GC 15 |
48.5 (5.0) |
50.1 (4.8) |
0.135 |
|
GC 16 |
43.3 (5.2) |
44.3 (4.6) |
0.368 |
|
Superior GC |
345.6 (39.9) |
355.4 (37.3) |
0.231 |
|
Inferior GC |
351.8 (36.3) |
362.6 (36.3) |
0.194 |
|
Temporal GC |
339.9 (37.8) |
352.9 (36.3) |
0.188 |
|
Nasal GC |
357.5 (38.8) |
365.1 (36.5) |
0.303 |
|
Total GC |
697.4 (74.9) |
718.1 (71.5) |
0.180 |
Statistics, the authors have uses Mann-Whitney test. Could they elaborate on why? Taking into account the number of measurements made, this reviewer thinks that other statistical tests should have been used. As long there is a clear reason behind it.
Although the number of eyes included is high, we conducted a 1-sample Kolmogorov-Smirnov test to verify the normality of the data distribution. However, ONL in temporal region did not follow a normal distribution, so non-parametric tests were used to prevent statistical bias. We modified the manuscript to explain this fact:
“A 1-sample Kolmogorov-Smirnov test was used to verify the normality of the data distribution. As ONL in temporal region did not follow a normal distribution, non-parametric tests were used.” (page 5 lines 4-5)
MINOR COMMENTS
Rephrase sentence in line 5 page 2. It is weird that they indicate that the authors hypothesized that the GC-IPL thinning could be an early indicator, but then the authors refer to a paper that does not belong to them. I assume they aimed to say that other researchers... also in the way the sentence is written it should read "made the authors hypothesize" (without the d).
We apologize and we rephrase the sentence to: “recently results made other researchers hypothesize that GCIPL thinning could be an early indicator of it in screening use (11).” (page 2, lines 7-9)
The degree symbol is not correct.
We changed it to: “which is referred as GCIPL in literature” and all along the text.
page 2 , line 6)
Last but not least, it is surprising that authors do not have any financial support (not even internal) nor acknowledgments.
We agree with the reviewer. Actually, we are searching for investment. However, we don’t have support today, so we have to pay the fee of the manuscript by ourselves.
Reviewer 3 Report
In this study Casado and colleagues found an association between outer nuclear layer (ONL) thicknesses and hydroxychloroquine (HCQ) retinopathy. In particular they correlate visual field (VF) damage and duration of the treatment with ONL thinning by using an objective analysis to assess these changes.
The manuscript is well write, and the data are well organized. The results are convincing and the authors discussed both the advantages and disadvantages of the experimental approach used and the results obtained.
Minor concern:
What is striking is that the enlisted population is predominantly women. Is there an explanation or is it causal? Is it possible to make qualitative-quantitative evaluations of the results obtained related to gender?
Author Response
REVIEWER 3
In this study Casado and colleagues found an association between outer nuclear layer (ONL) thicknesses and hydroxychloroquine (HCQ) retinopathy. In particular they correlate visual field (VF) damage and duration of the treatment with ONL thinning by using an objective analysis to assess these changes.
The manuscript is well write, and the data are well organized. The results are convincing and the authors discussed both the advantages and disadvantages of the experimental approach used and the results obtained.
Minor concern:
What is striking is that the enlisted population is predominantly women. Is there an explanation or is it causal? Is it possible to make qualitative-quantitative evaluations of the results obtained related to gender?
Similarly to reviewer 2, this is a logical doubt. This is probably since the rheumatic diseases in which hydroxychloroquine treatment is used (Lupus, rheumatoid arthritis, ...) are much more frequent in females than in males (Yoo DH, Suh CH, Shim SC, Jeka S, Cons-Molina FF, Hrycaj P, et al. A multicentre randomised controlled trial to compare the pharmacokinetics, efficacy and safety of CT-P10 and innovator rituximab in patients with rheumatoid arthritis. Ann Rheum Dis. 2017 Mar;76(3):566-570.). Similarly, HCQ is much more frequent in females. For instance, in reference 12 (de Sisternes L, Hu J, Rubin DL, Marmor MF. Localization of damage in progressive hydroxychloroquine retinopathy on and off the drug: inner versus outer retina, parafovea versus peripheral fovea. Invest Ophthalmol Vis Sci 2015;56(5):3415–3426.) de Sisternes et al included 11 patients with HCQ, and 10 of them were females.
Round 2
Reviewer 2 Report
The authors of the paper “Outer nuclear layer damage for detection of early retinal toxicity of hydroxychloroquine” describe a series of measurements performed with OCT to discern whether they can detect early GCQ retinal toxicity by measuring ONL.
The authors have improved the paper considerably after this revision, and now it reads and results are interpreted easily. They have also addressed the questions raised by the reviewers properly.
Minor comments:
- This reviewer assumes that when an eye was removed from the study, the patient was not. Could this be confirmed? The reason of the question comes from line 28 page 2: 124 eyes from 81 individuals (75F and 6M). In line 36: 10 eyes were removed (5F and 1M). This then should leave it in 70F and 5M (=75 individuals). In page 5 lines 12 then it is indicated 114 eyes of 76 individuals (71F and 5M). If an individual was included for one eye but not the other, can authors find a way to briefly indicate that and avoid confusion?
- Page 5 line 16: Authors start a sentence with number 55. Please write the number with letters.
- Page 6 line 4: I think sentence should read “In Table 4 are shown the …”. If authors find it appropriate, please change.
- Page 7 line 29: Degree symbol was not corrected. It reads 3º x 3º instead of 3° x 3°
Author Response
- This reviewer assumes that when an eye was removed from the study, the patient was not. Could this be confirmed? The reason of the question comes from line 28 page 2: 124 eyes from 81 individuals (75F and 6M). In line 36: 10 eyes were removed (5F and 1M). This then should leave it in 70F and 5M (=75 individuals). In page 5 lines 12 then it is indicated 114 eyes of 76 individuals (71F and 5M). If an individual was included for one eye but not the other, can authors find a way to briefly indicate that and avoid confusion?
We agree with the reviewer, so we added the following sentence: "
If the fellow eye might be used in this study, only the affected eye was removed from the study, but the patient was not.
- Page 5 line 16: Authors start a sentence with number 55. Please write the number with letters.
We agree and we change this.
- Page 6 line 4: I think sentence should read “In Table 4 are shown the …”. If authors find it appropriate, please change.
Similarly, we agree and we change this.
Page 7 line 29: Degree symbol was not corrected. It reads 3º x 3º instead of 3° x 3°
Finally, we also change this.